# Standard echocardiography versus handheld echocardiography for the detection of subclinical rheumatic heart disease: a systematic review and meta-analysis of diagnostic accuracy

Lisa Helen Telford [iD],[1,2] Leila Hussein Abdullahi,[2,3] Eleanor Atieno Ochodo,[4] Liesl Joanna Zuhlke [iD],[5,6] Mark Emmanuel Engel[1]

► Prepublication history and supplementary material for this paper is available online. To view these files, please visit the journal online (http://dx.doi.org/10.1136/bmjopen-2020-038449).

For numbered affiliations see end of article.

**Correspondence to**
Lisa Helen Telford;
lisa.telford@uct.ac.za

## ABSTRACT

**Objective** To summarise the accuracy of handheld echocardiography (HAND) which, if shown to be sufficiently similar to that of standard echocardiography (STAND), could usher in a new age of rheumatic heart disease (RHD) screening in endemic areas.

**Design** Systematic review and meta-analysis.

**Data sources** PubMed, Scopus, EBSCOHost and ISI Web of Science were initially searched on 27 September 2017 and again on 3 March 2020 for studies published from 2012 onwards.

**Eligibility criteria** Studies assessing the accuracy of HAND compared with STAND when performed by an experienced cardiologist in conjunction with the 2012 World Heart Federation criteria among populations of children and adolescents living in endemic areas were included.

**Data extraction and synthesis** Two reviewers independently extracted data and assessed the methodological quality of included studies against review-specific Quality Assessment of Diagnostic Accuracy Studies (QUADAS)-2 criteria. A meta-analysis using the hierarchical summary receiver operating characteristic model was conducted to produce summary results of sensitivity and specificity. Forest plots and scatter plots in receiver operating characteristic space in combination with subgroup analyses were used to investigate heterogeneity. Publication bias was not investigated.

**Results** Six studies (N=4208) were included in the analysis. For any RHD detection, the pooled results from six studies were as follows: sensitivity: 81.56% (95% CI 76.52% to 86.61%) and specificity: 89.75% (84.48% to 95.01%). Meta-analytical results from five of the six included studies were as follows: sensitivity: 91.06% (80.46% to 100%) and specificity: 91.96% (85.57% to 98.36%) for the detection of definite RHD only and sensitivity: 62.01% (31.80% to 92.22%) and specificity: 82.33% (65.15% to 99.52%) for the detection of borderline RHD only.

**Conclusions** HAND displayed good accuracy for detecting definite RHD only and modest accuracy for detecting any RHD but demonstrated poor accuracy for the detection of borderline RHD alone. Findings from this review provide

### Strengths and limitations of this study

► Language restrictions were not imposed during the literature search to minimise the chance of missing studies.
► Data extraction was performed by two independent reviewers, thereby reducing the risk of bias.
► Insufficient reporting limited our ability to adequately assess risk of bias and investigate potential sources of heterogeneity.
► The small number of included studies prevented us from performing meta-regression.

some evidence for the potential of HAND to increase access to echocardiographic screening for RHD in resource-limited and remote settings; however, further research into feasibility and cost-effectiveness of wide-scale screening is still needed.

**PROSPERO registration number** CRD42016051261.

## INTRODUCTION

Rheumatic heart disease (RHD) is an acquired permanent heart valve condition which results from an atypical immune reaction to group A streptococcal infection typically occurring in childhood.[1 2] Disease progression leading to chronic RHD can result in irreversible heart valve damage, cardiac failure and premature death.[3 4] RHD is, however, a preventable and treatable chronic condition which most often effects disadvantaged populations.[3 5]

Significantly, RHD can remain asymptomatic for many years, particularly during the initial stages, thereby hindering the timely implementation of penicillin prophylaxis.[6] Echocardiographic screening to identify those with subclinical disease has been advocated as a means to support secondary prevention and potentially slow disease progression to overt clinical RHD.[7 8] Yet the feasibility of wide-scale

echocardiographic screening remains hindered by high costs and the scarcity of trained personnel.[9] Alternative RHD screening tests, which are both accurate and affordable, are therefore needed in many endemic areas.

Handheld echocardiography (HAND) is a non-invasive, highly portable and comparatively less expensive device which has been presented in recent publications to be a promising alternative to standard echocardiography (STAND), despite some limitations such as a lack of spectral Doppler capabilities.[10 11] For HAND to be considered a suitable replacement for STAND, the device's accuracy needs to be similar to that of STAND.

We conducted a systematic review and meta-analysis of studies assessing the diagnostic accuracy of HAND for the detection of RHD in children and adolescents. The findings of this review may offer direction to guideline developers as well as assist with the identification of gaps in diagnostic testing for RHD in endemic areas.

## METHODS
This systematic review was prepared according to the Preferred Reporting Items for a Systematic Review and Meta-Analysis of Diagnostic Test Accuracy Studies (PRISMA-DTA) guidelines.[12] The protocol for this review is registered with the International Prospective Register of Systematic Reviews (PROSPERO) under the registration number CRD42016051261 and has been published in *BMJ Open*.[13]

### Patient and public involvement
No patient involved.

### Data sources and study eligibility
Studies were considered eligible for inclusion if the following criteria were met: (1) the accuracy of HAND compared with STAND when performed by an experienced cardiologist and in conjunction with the 2012 World Heart Federation (WHF) criteria was evaluated, and (2) the sample consisted of populations of children and adolescents living in endemic areas. Only primary observational studies of either a cross-sectional, cohort or diagnostic case–control design were considered. Descriptive studies such case studies and case series were excluded as were studies reporting on the same data. Studies using non-handheld devices as the index test or criteria other than the 2012 WHF criteria in combination with STAND as the reference test were also excluded.

We conducted systematic electronic literature searches of four sources (PubMed, Scopus, EBSCOHost and ISI Web of Science) using predefined tailor-made strategies (see online supplemental file 1). No restrictions in terms of language were applied; however, searches were limited to articles published from 2012 onwards. Both published and unpublished literature were considered eligible for inclusion. A manual search of the reference lists of all included studies as well as relevant review articles was also conducted.

The titles and/or abstracts of all identified articles were screened independently by two reviewers. During this process, and on the basis of predefined eligibility criteria, all clearly ineligible studies were excluded. Discrepancies regarding eligibility were resolved through discussion and consensus. Some authors were contacted for additional information on published data.

### Data extraction and management
Using a predefined data extraction form, two reviewers independently extracted information on metrics of diagnostic accuracy: numbers of true positives (TP), false positives (FP), true negatives (TN) and false negatives (FN) as well as other covariates relating to study characteristics, population, reference and index test details, test outcome and number of missing or unavailable test results from all included studies.

For accuracy measures, sensitivity and specificity were calculated using the numbers of TP, FP, TN and FN in accordance with standard convention. Data extraction conflicts were resolved through discussion and with the assistance of a third reviewer where necessary. Information garnered through the data extraction process was used to determine each study's quality as well as for synthesising evidence.

### Assessment of methodological quality
A review-specific Quality Assessment of Diagnostic Accuracy Studies (QUADAS)-2 tool was used to assess the risk of bias and concerns regarding applicability in all included studies.[14] The tool, encompassing four domains, was tailored to meet the specific requirements of this review. Two reviewers independently assessed the risk of bias in all included studies according to review-specific QUADAS-2 criteria. Discrepancies were resolved through discussion until consensus was reached and the assistance of a third reviewer was enlisted when necessary.

### Statistical analysis and data synthesis
A meta-analysis using the hierarchical summary receiver operating characteristic (HSROC) model was conducted to produce summary results of sensitivity and specificity. The HSROC model was used for meta-analysis as it accounts for variations in test thresholds.

Statistical measures of variability or heterogeneity to determine whether or not to conduct a meta-analysis were not used. While the Cochrane Q test and $I^2$ are routinely used to examine and quantify the amount of variability in meta-analyses of intervention studies where there is a single measure of effect or univariate outcome such as a risk or OR,[15 16] meta-analyses of diagnostic test accuracy generally do not employ such measures.[17] Instead, forest plots and scatter plots in receiver operating characteristic (ROC) space in combination with subgroup analyses were used to investigate heterogeneity.

Data were analysed according to three categorisations of RHD: any RHD (definite or borderline), definite RHD only and borderline RHD only. The any RHD category

was selected as the main meta-analysis as it had the most complete data. We were unable to extract metrics of diagnostic accuracy for the definite and borderline RHD only categories from Beaton (2016) and therefore excluded this study from these meta-analyses. We chose to use nurse A's results for Mirabel (2015) since nurse A and nurse B both interpreted the same HAND images which prevented the pooling of data.

Data from Zühlke (2016) were included in the analysis and synthesis of data even though the age range of participants fell outside the predefined range for eligibility. It was determined that this study should be included, regardless, since the data overall were quite few and the variation in age was not significant enough to warrant exclusion. However, data from Zühlke (2016) were excluded from all summary estimates of disease prevalence since this study used a nested case–control design which predetermines disease prevalence by design.

Heterogeneity was examined for the main meta-analysis only. We were only able to investigate the relationship between test accuracy and echocardiographer expertise through subgroup analysis. A sensitivity analysis was performed instead of subgroup analysis for the categorical covariates; HAND protocol and geographical location due to the skewed distribution of studies within each subgroup. We were unable to perform meta-regression for the covariates; age and sex due to insufficient and inadequately reported data.[17] We were also unable to conduct a sensitivity analysis on risk of bias since no studies were found to have a high risk of bias. All plots were generated using the Review Manager (RevMan) software package, V.5.3.[18] Meta-analysis was performed using SAS software, V.9.4.[19]

We did not investigate publication bias as methods of assessing publication bias for studies of diagnostic accuracy are still being developed. While the Deeks test has been suggested for use in diagnostic accuracy studies, the test has low power for detecting asymmetry in funnel plots, particularly when a large amount of heterogeneity is present.[17]

## RESULTS
### Results of the search
Results of the literature search are reported in accordance with the PRISMA statement and the study selection process is illustrated in figure 1.[20] All electronic searches were performed by two independent reviewers on 27 September 2017. Combined, the search yielded a total of 92 records, of which 9 were duplicates. A total of 67 were excluded based on title or abstract, leaving 16 for full-text review. Nine studies were excluded on the basis of a full-text review, while an additional study was also excluded after consultation with study authors. Six studies which met the predefined eligibility criteria were included in this review.

The same search was re-run on 3 March 2020 to check for any additional eligible studies. Only one potentially eligible study[21 22] was found but has been excluded on the basis of being an abstract-only publication with no full-text available for review.

### Included studies
A summary of notable characteristics of all included studies[23–28] is shown in table 1. One study did not avoid a case–control design, however, cases and controls were sampled from the same population. Research has shown that case–control studies that use alternative diagnosis controls, controls from non-endemic areas or confirmed disease-free (healthy) controls tend to overestimate specificity.[29]

Significantly, all but two studies were conducted in Africa. Screening was performed in RHD endemic areas among children and adolescents with most studies being school based. Combined, all six studies included a total of 4208 participants, of which 54% were female. The pooled mean age of participant's was 10.8 years (SD: ±1.9).

All included studies used the same make of handheld device; the Vscan machine (General Electric, Medical Systems, Milwaukee, Wisconsin, USA) paired with a 1.7 to 3.4 MHz transducer. These machines provide both two-dimensional (2D) and colour imaging on an integrated 8.9 cm display.[24 26] Frame rates range from 25 to 30 Hz for greyscale imaging and from 12 to 16 Hz for colour Doppler.[23 27] Vscan machines are, however, limited by a lack of spectral Doppler capabilities.[24]

### Excluded studies
Ten studies[9 10 30–37] were excluded during full-text screening. Reasons for exclusion included abstract-only publications,[32 33 35] the use of ineligible reference[30 31] or index[34] tests, the use of duplicate data,[10 37] not specifying the test threshold a priori[9] and not being a study of diagnostic accuracy.[36]

### Methodological quality of included studies
A summary of the assessment of methodological quality of all included studies is illustrated in figure 2. Overall, only two[25 26] of the six[23–28] included studies were assessed as having a low risk of bias, while the risk in the remaining four[23 24 27 28] was unclear. Two studies[23 28] had participant selection bias concerns. Of these, both[23 28] failed to adequately describe participant enrolment methods, while one[28] also did not avoid a case–control design. The risk of bias in terms of flow and timing was unclear in two studies.[24 27] Of these, one[27] did not include all participants in the analysis due to technical difficulties, while the time interval between the index and reference test was unclear in the other.[24] Overall, time intervals between index and reference tests were poorly described. Likewise, reporting of quality control of the index test was uniformly poor across all included studies. Concerns regarding applicability were low in all six[23–28] studies.

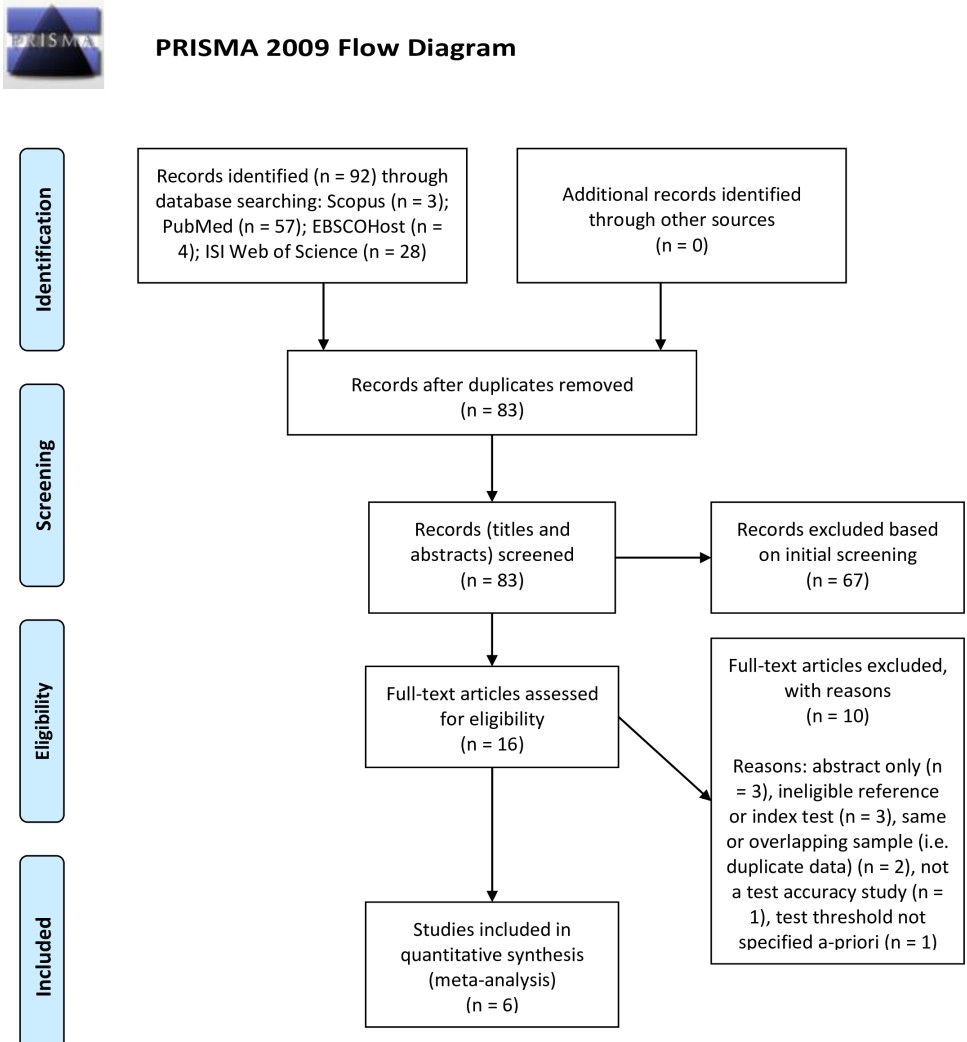

**Figure 1** Preferred Reporting Items for a Systematic Review and Meta-Analysis (PRISMA) flow diagram illustrating study identification, selection, eligibility and inclusion. From Moher *et al.*[20]

## Findings

### For any RHD

A total of six evaluations of HAND for any RHD were performed with data from six studies and a total of 4208 participants. Pooled prevalence of any RHD from five included studies was 12% (95% CI 6% to 19%). The forest plot revealed little variation in estimates of sensitivity and specificity and the HSROC plot (see online supplemental file 2 for all plots) revealed moderate accuracy of the test. Meta-analytical sensitivity and specificity (95% CI) of data at mixed thresholds were 81.56% (76.52% to 86.61%) and 89.75% (84.48% to 95.01%), respectively.

### For definite RHD

A total of five evaluations of HAND for definite RHD were performed with data from five studies and a total of 3588 participants. Pooled prevalence of definite RHD from four included studies was 6% (95% CI 2% to 12%). The forest plot revealed some variation in estimates of specificity while estimates of sensitivity were largely homogeneous, with the exception of a single outlier. The HSROC

plot indicated good accuracy of the test. Meta-analytical sensitivity and specificity (95% CI) of data at mixed thresholds were 91.06% (80.46% to 100%) and 91.96% (85.57% to 98.36%), respectively.

### For borderline RHD

A total of five evaluations of HAND for borderline RHD were performed with data from five studies and a total of 3685 participants. Pooled prevalence of borderline RHD from four included studies was 20% (95% CI 6% to 39%). The forest plot revealed some variation in estimates of specificity, while estimates of sensitivity were largely homogeneous with the exception of a single outlier. The HSROC plot indicated poor accuracy of the test. Meta-analytical sensitivity and specificity (95% CI) of data at mixed thresholds were 62.01% (31.80% to 92.22%) and 82.33% (65.15% to 99.52%), respectively.

### Investigations of heterogeneity

Heterogeneity or variation between studies was investigated both visually and through subgroup analysis for the

**Table 1**  Summary of characteristics of included studies: ordered alphabetically by study author

| Study | Study design | City, country (Classification*) | Recruitment site/setting | Participant selection method | Sample size (N) | % Female | Mean age (years) | SD |
|---|---|---|---|---|---|---|---|---|
| Beaton, 2014[23] | Spiked cohort | Kampala, Uganda (low income) | A school and the Mulago Hospital Complex | Unclear | 125 | 55.2 | 10.8† | – |
| Beaton, 2015[24] | Cross sectional | Gulu, Uganda (low income) | 5 public schools | A random 10% subset of the entire sample plus any child with mitral or aortic regurgitation were preselected to receive HAND | 1420 | 53 | 10.8 | ±2.6 |
| Beaton, 2016[25] | Cross sectional | Belo Horizonte, Brazil (upper-middle income) | 2 primary and 3 secondary public schools | A subset of the sample containing all STAND abnormals plus a random 25% of all STAND normals were preselected for HAND | 397 | 49.1 | 13.9 | ±2.6 |
| Mirabel, 2015[26] | Cross sectional | Nouméa, New Caledonia (high income) | Primary schools | Consecutive | 1217 | 50.5 | 9.6 | ±0.5 |
| Ploutz, 2016[27] | Cross sectional | Gulu, Uganda (low income) | 2 primary schools | Consecutive | 956 | 60.7 | 11.1 | ±2.5 |
| Zühlke, 2016[28] | Nested case–control | Cape Town, South Africa (upper-middle income) | Schools | Unclear | 93 | 68.8 | 17†‡ | – |
| Total | | | | | 4208 | 54 | 10.8 | ±1.9 |

*According to the World Bank's economic classification system.
†Excluded from pooled mean and SD calculations (incomplete or incomparable data).
‡Median age (mean age not available).
HAND, handheld echocardiography; STAND, standard echocardiography.

main meta-analysis only. We were only able to perform this analysis for the any RHD category as the data were too few to enable model convergence for the definite and borderline RHD only categories.

### Covariates in the models
We were only able to use one of the five prespecified covariates to investigate heterogeneity due to insufficient and inadequately reported data. HAND echocardiographer expertise (expert vs non-expert) was investigated as a possible source of heterogeneity through subgroup analysis. Half of all included studies evaluated the accuracy of HAND when performed and interpreted by trained non-experts, while the other half assessed its accuracy in the hands of experts.

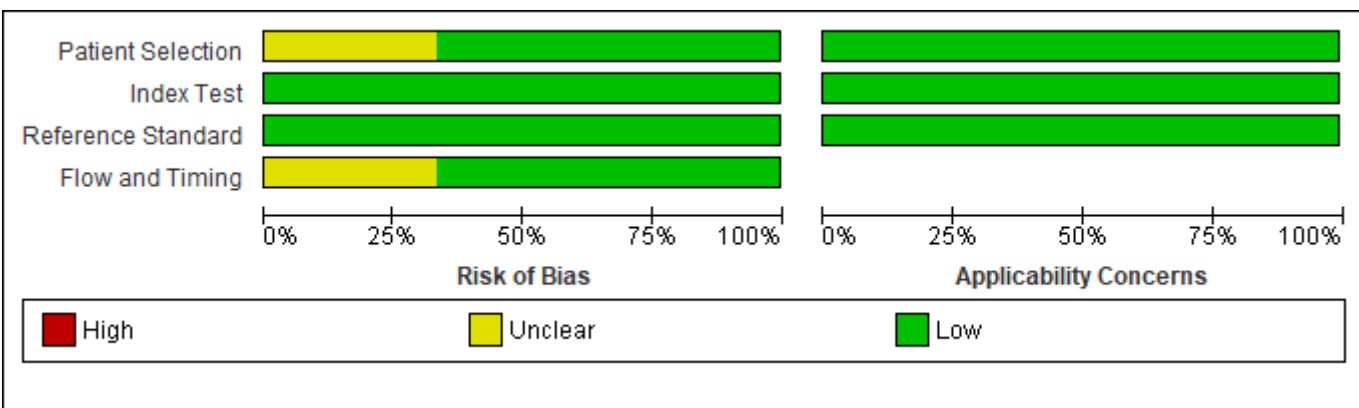

**Figure 2**  Risk of bias and applicability concerns graph: review authors' judgements about each domain presented as percentages across included studies.

**Table 2** Sources of heterogeneity for handheld echocardiography for any RHD

| Group | Covariate | Subgroup | n (N=6) | Median pooled sensitivity (95% Crl) | Median pooled specificity (95% Crl) |
|---|---|---|---|---|---|
| **Overall** | | | 6 | **81.56% (76.52–86.61)** | **89.75% (84.48–95.01)** |
| **Subgroup analysis** | HAND interpreter expertise | Expert | 3 | 82.54% (74.71–90.37) | 94.57% (87.52–100) |
| | | Non-expert | 3 | 80.76% (73.18–88.33) | 85.71% (79.91–91.51) |
| **Sensitivity analysis** | Geographical location | Overall | | 80.40% (64.96–95.84) | 87.15% (76.35–97.96) |
| | | Low-income and middle-income countries | 5 | 81.17% (75.63–86.71) | 90.07% (82.98–97.15) |
| | HAND protocol | Overall | | 85.35% (80.60–90.09) | 88.80% (84.55–93.06) |
| | | Multiple views | 5 | 80.98% (75.38–86.59) | 87.46% (83.24–91.67) |

HAND, handheld echocardiography; Crl, credible interval; n, number of studies; N, total number of included studies.

## Subgroup and sensitivity analyses

A subgroup analysis was performed to investigate variations in echocardiographer expertise as a potential source of heterogeneity. Since no studies were found to have a high risk of bias, we did not explore the effect of excluding such studies on the accuracy of summary estimates. Sensitivity analyses were, however, conducted to investigate the effect of removing a single study on summary estimates of sensitivity and specificity for the covariates: geographical location and HAND protocol.

A subgroup analysis for the covariate; echocardiographer expertise, as shown in table 2 and illustrated in online supplemental file 3, revealed that both sensitivity (82.54% vs 80.76%) and specificity (94.57% vs 85.71%) were higher for any RHD detection using HAND when tests were performed and interpreted by experts compared with non-experts.

Sensitivity analyses (see table 2) were performed to investigate the effect of excluding (1) the single high-income country study and (2) the study which employed a single view protocol on the accuracy of summary estimates. We found that both sensitivity (81.17% vs 80.4%) and specificity (90.07% vs 87.15%) increased compared with the overall analysis when only low-income and middle-income country studies were considered whereas both sensitivity (80.98% vs 85.35%) and specificity (87.46% vs 88.8%) decreased in comparison with the overall analysis when only studies which employed multiple view protocols were considered.

## DISCUSSION
## Summary of main findings

We evaluated the accuracy of HAND for three distinct disease categories and found that, overall, the test was both sensitive and specific for detecting definite RHD only and moderately accurate for detecting any RHD but demonstrated insufficient accuracy for detecting borderline RHD alone.

Findings from this review provide some evidence for the potential of HAND to increase access to echocardiographic screening for RHD in resource-limited and remote settings. A summary of the accuracy estimates produced by meta-analysis using the HSROC method is included in table 3.

## Strengths and limitations of this review
### Strengths

We have evaluated and summarised the accuracy of HAND for the detection of RHD in endemic areas, making this review relevant to current global agendas. This review also serves to highlight the existing gaps in evidence for which further research could be beneficial. We did not impose any limits in terms of language during the literature search so as to minimise the chance of missing studies. Data extraction was performed by two independent reviewers, thereby reducing the risk of bias.

### Limitations

There were a number of shortcomings of this review, which included the following.

#### Eligibility

We were unable to include studies which used STAND in conjunction with criteria other than the 2012 WHF criteria as the reference standard, which limited the number of studies eligible for inclusion.

#### Quality of included studies

Insufficient reporting of participant characteristics and study methods including study design, participant selection and test timing restricted our ability to adequately assess risk of bias and investigate potential sources of heterogeneity.

#### Paucity of data

Insufficient and inadequately reported data, as well as the presentation of aggregate data, limited the scope of our investigations of heterogeneity, while the small number

**Table 3** Summary of findings

*What is the diagnostic accuracy of handheld echocardiography in detecting any RHD (definite or borderline)?*

| | |
|---|---|
| **Patients/Population** | People residing in areas endemic for RHD (6 out of 6 studies) |
| **Prior testing with echo** | Yes (2 studies), no (4 studies) |
| **Settings** | 5 out of 6 screening studies were field setting (communities and schools) based, while 1 study was half hospital registry follow-up, half school based. 4 of the 6 studies were conducted in Africa with 3 of those from Uganda. |
| **Index test(s)** | General Electric (GE) Vscan handheld machine (6 out of 6 studies) |
| **Reference standard** | Standard echocardiography (2D, continuous-wave, and colour-Doppler echocardiography) performed by an experienced imager and in conjunction with the 2012 WHF criteria (6 out of 6 studies). Test brands included GE Vivid-I ultrasound machine (2 studies); GE Vivid-Q ultrasound machine (2 studies); Philips CX-50 ultrasound machine (1 study); either a GE Vivid-I or Q or Philips CX-50 ultrasound machine (1 study) |
| **Importance** | HAND is being used as first line replacement for STAND in disease screening programmes for RHD, as it is comparably inexpensive, quick, user friendly, easy to interpret and may have similar sensitivity to STAND |
| **Studies** | Cross-sectional (n=4), spiked cohort (n=1) and nested case–control (n=1) studies. More than half (n=4) of all included studies did not explicitly state the study design used and were thus assigned a study design based on other reported characteristics and participant enrolment methods used. |
| **Quality concerns** | Poor reporting of study design, participant characteristics and pretest probability were common concerns. For the majority of studies the risk of bias was unclear in terms of 'patient selection' and 'flow and timing'. Concerns regarding applicability were low in all included studies. |

| Test types | Number of participants* (n) | Summary estimates (95% credible CI) |
|---|---|---|
| HAND for **any RHD** | 4208 (6) | Sensitivity: 81.56% (76.52–86.61) Specificity: 89.75% (84.48–95.01) |
| HAND for **definite RHD** | 3588 (5) | Sensitivity: 91.06% (80.46–100) Specificity: 91.96% (85.57–98.36) |
| HAND for **borderline RHD** | 3685 (5) | Sensitivity: 62.01% (31.8–92.22) Specificity: 82.33% (65.15–99.52) |

*Excluding participants with other diagnoses on STAND.

.HAND, handheld echocardiography; n, number of studies; RHD, rheumatic heart disease; STAND, standard echocardiography.

of included studies prevented us from performing meta-regression. Overall, the findings from this review may lack power due to the small sample size.

### Applicability of findings to the review question

Concerns regarding the applicability of included studies to the review question were considered low according to review-specific QUADAS-2 criteria. Since all but one were conducted in low- or middle-income countries, and all studies with one exception were conducted in field settings, the results of this review are applicable for use in endemic areas for which screening programmes are frequently targeted. However, our limited assessment of risk of bias and investigations into sources of heterogeneity such as age and gender due to insufficient reporting may lessen the applicability of findings to the review question.

In the context of disease control programmes, being able to demonstrate variation in test accuracy associated with factors such as age and gender would be beneficial for policy makers. Fully understanding included studies'

risk of bias would also assist in objectively assessing the strength of evidence. For these reasons, prospective authors of diagnostic test accuracy studies are urged to make use of the Standards for Reporting of Diagnostic Accuracy Studies (STARD) guidelines[38] when reporting methods of study design and conduct.

### CONCLUSION

This review provides a summary of the accuracy of HAND for the detection of RHD. In populations of children and adolescents living in RHD endemic areas, HAND is both sensitive and specific for detecting definite RHD. The device is less accurate at detecting any RHD and demonstrates substandard accuracy for the detection of borderline RHD only. Nonetheless, HAND may hold value as a replacement for first-line screening due to its high sensitivity for definite RHD detection and adequate accuracy for any RHD detection.

## Implications for practice

We have summarised the accuracy of HAND when used as a screening tool; however, the device's potential value in terms of diagnostics has yet to be established. We therefore posit that HAND could be recommended as an acceptable replacement test for first-line screening in endemic areas provided a standardised set of device-specific diagnostic criteria are developed.

Another key consideration is the applicability of these findings for recommendations to integrate screening into routine clinical practice. A recent publication has reviewed the cost-effectiveness of screening in high-risk populations[39] and determined that screening all indigenous Australian aged 5–12 years old in half of their communities in alternate years was found to be cost-effective, if RHD can be detected at least 2 years earlier. However, this result was sensitive to a number of assumptions, including local costs and context. Other cost-effectiveness models have also suggested modestly improved outcomes at lower cost.[40] Neither of these studies included the significant cost-reduction of using HAND instead of STAND, hence, we highly recommend adding a cost-effectiveness analysis into proposed new screening studies.

Finally, our findings demonstrate comparable results by non-experts; this has also been demonstrated in several other reports,[31 41] but again there are no detailed cost-effectiveness analyses using non-experts and HAND.

## Implications for research

The findings of this review highlight the need for a new set of evidence-based guidelines tailored to the capabilities of HAND in order to maximise the device's diagnostic potential. Further studies assessing the diagnostic accuracy of HAND when using a standardised protocol are needed as is further research into the feasibility, cost-effectiveness and consequences of implementing wide-scale screening programmes. Furthermore, the development of standardised training programmes for non-experts is recommended as screening for RHD in endemic areas inevitably rests on the success of task shifting.[27] We conclude that while HAND has been shown to be sufficiently accurate for the detection of RHD, there is still a need for further research before its wide-scale use can be endorsed.

**Author affiliations**
[1]Department of Medicine, Groote Schuur Hospital, University of Cape Town Faculty of Health Sciences, Cape Town, Western Cape, South Africa
[2]Department of Paediatrics and Child Health, Red Cross War Memorial Children's Hospital, University of Cape Town Faculty of Health Sciences, Cape Town, Western Cape, South Africa
[3]Department of Policy and Research, African Institute for Development Policy (AFIDEP), Nairobi, Kenya
[4]Department of Global Health, Faculty of Medicine and Health Sciences, University of Stellenbosch Centre for Evidence-Based Health Care, Cape Town, Western Cape, South Africa
[5]Division of Cardiology, Department of Paediatrics and Child Health, Red Cross War Memorial Children's Hospital, University of Cape Town Faculty of Health Sciences, Cape Town, Western Cape, South Africa
[6]Division of Cardiology, Department of Medicine, Groote Schuur Hospital, University of Cape Town Faculty of Health Sciences, Cape Town, Western Cape, South Africa

**Contributors** LJZ and MEE jointly conceived of the study and provided content advice. LHT and LHA performed the systematic review of the literature, independently extracted the data and are responsible for the quality control of study selection. EAO provided methodological advice and was responsible for conducting the meta-analysis. LHT significantly contributed to the interpretation of the data and wrote the first draft of the manuscript. All authors contributed to editing subsequent versions of the manuscript.

**Funding** LJZ, LHT and LHA all receive funding from the Medtronic Foundation through support to RHD Action. LHT receives support from the National Research Foundation of South Africa (NRFSA) through its internship programme. LJZ is supported by both the NRFSA and Medical Research Council (MRC).

**Competing interests** None declared.

**Patient consent for publication** Not required.

**Provenance and peer review** Not commissioned; externally peer reviewed.

**Data availability statement** All data relevant to the study are included in the article or uploaded as supplementary information.

**ORCID iDs**
Lisa Helen Telford http://orcid.org/0000-0002-8554-057X
Liesl Joanna Zuhlke http://orcid.org/0000-0003-3961-2760

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
