## [Reviewer comments · BMJ Open]

ARTICLE DETAILS

TITLE (PROVISIONAL)	Standard echocardiography versus handheld echocardiography for the detection of subclinical rheumatic heart disease: A systematic review and meta-analysis of diagnostic accuracy
AUTHORS	Telford, Lisa; Abdullahi, Leila; Ochodo, Eleanor; ZUHLKE, LIESL; Engel, Mark

VERSION 1 – REVIEW

REVIEWER	AKM Monwarul Islam National Institute of Cardiovascular Diseases, Dhaka, Bangladesh
REVIEW RETURNED	26-Mar-2020

GENERAL COMMENTS	Good work. However, there are few observations. 1. In the METHODS section, 'No restrictions in terms of language were applied, however, searches were limited to articles published from 2012 onwards.'; the exact duration of literature search should be mentioned. 2. In Table 3, 'Prior testing with echo' has been reported Yes in 2 studies, and No in 5 studies, but the total number of studies finally included in this metaanalysis is 6; what is the explanation?
--

REVIEWER	Bruno Ramos Nascimento Hospital das Clínicas da Universidade Federal de Minas Gerais
REVIEW RETURNED	28-Mar-2020

GENERAL COMMENTS	Dear authors, Congratulations for putting together a meta-analysis of substantial importance for RHD research, addressing a question especially relevant for low-income settings, where personnel and equipment are scarce. The meta-analysis was well conducted, and the results clearly presented. However, I have a few minor questions to be addressed in order to make the manuscript more comprehensive. Please see the points below: - For better understanding, the table with detailed items (points) of the QUADAS-2 quality score should be provided. The final score should be added to Table 1, in a separate column, to provide an overview of study quality. - Even in the absence of studies with low quality, the subgroup analysis excluding the 2 studies with lower quality should be provided (can be only in the text, without necessarily being included in the summary table), for completeness. - The summary ROC curves would be illustrative, as an appendix figure. It's a visual approach for accuracy, provided its area and 95% CIs.
---

	- The authors did not report heterogeneity measures, nor the inconsistency report I² (I-squared) for pooled measures. Please provide these. - For guiding decision, Likelihood ratios with respective 95% CI should be provided (any RHD and definite RHD).
--	--

VERSION 1 – AUTHOR RESPONSE

Comments from Reviewer 1	Responses
Good work. However, there are few observations:	Thank you.
1. In the METHODS section, 'No restrictions in terms of language were applied, however, searches were limited to articles published from 2012 onwards.'; the exact duration of literature search should be mentioned.	We believe we have addressed this in the 'results of the search' section on page 9 as follows: "All electronic searches were performed by two independent reviewers on September 27th, 2017" and on page 10 as follows: "The same search was re-run on March 3rd, 2020 to check for any additional eligible studies."
2. In Table 3, 'Prior testing with echo' has been reported Yes in 2 studies, and No in 5 studies, but the total number of studies finally included in this meta-analysis is 6; what is the explanation?	Thank you for noticing this. We had initially included 7 studies but had to later exclude one based on new evidence that it included the same data as another included study. Table 3 has been amended to reflect the 6 included studies only.
Comments from Reviewer 2	Responses
Dear authors, Congratulations for putting together a meta-analysis of substantial importance for RHD research, addressing a question especially relevant for low-income settings, where personnel and equipment are scarce. The meta-analysis was well conducted, and the results clearly presented. However, I have a few minor questions to be addressed in order to make the manuscript more comprehensive. Please see the points below:	Thank you for these suggestions.
1. For better understanding, the table with detailed items (points) of the QUADAS-2 quality score should be provided. The final score should be added to Table 1, in a separate column, to provide an overview of study quality.	Thank you for this point. We have added the QUADAS-2 figure as suggested. Thank you for this suggestion however we feel that the inclusion of the risk of bias and applicability concerns graph which summarizes the overall methodological quality of included studies is sufficient. We have, however, included further detailed referencing to reflect which studies received which overall

	quality score and why in the 'methodological quality of included studies' section on page 13.
2. Even in the absence of studies with low quality, the subgroup analysis excluding the 2 studies with lower quality should be provided (can be only in the text, without necessarily being included in the summary table), for completeness.	Thank you for this suggestion however if we were to exclude the studies with an unclear risk of bias (i.e. the lower quality studies) we would only be left with 2 studies with a low risk of bias. It is not possible to do a subgroup analysis for only 2 studies as the statistical models require at least 4 studies to converge.
3. The summary ROC curves would be illustrative, as an appendix figure. It's a visual approach for accuracy, provided its area and 95% CIs.	Thank you for this comment. Due to limitations on the overall number of tables and figures we elected to omit the SROC curves from the manuscript but have now included them as a supplementary file. Please see the supplementary file at the end of this letter.
4. The authors did not report heterogeneity measures, nor the inconsistency report I ² (I-squared) for pooled measures. Please provide these.	Thank you for noting this. We maintain that heterogeneity measures are recommended for intervention reviews but not for diagnostic test accuracy reviews such as this. We have therefore added the following paragraph to the manuscript in the methods section on page 8 justifying this: "Statistical measures of variability or heterogeneity to determine whether or not to conduct a meta-analysis were not used. While the Cochrane Q test and I ² are routinely used to examine and quantify the amount of variability in meta-analyses of intervention studies where there is a single measure of effect or univariate outcome such as a risk or odds ratio,[15][16] meta-analyses of diagnostic test accuracy generally do not employ such measures.[17] Instead, forest plots and scatter plots in Receiver Operating Characteristic (ROC) space in combination with subgroup analyses were used to investigate heterogeneity."
5. For guiding decision, Likelihood ratios with respective 95% CI should be provided (any RHD and definite RHD).	Thank you for this suggestion. However, we contend that reporting likelihood ratios is not mandatory. We have instead provided pooled estimates of sensitivity and specificity as it was determined that these measures would be sufficient. While we acknowledge that likelihood ratios are useful in guiding clinical decisions, the findings from our study are more geared towards informing screening practices and disease detection as opposed to clinical guidance.

VERSION 2 – REVIEW

REVIEWER	AKM Monwarul Islam National Institute of Cardiovascular Diseases, Bangladesh
REVIEW RETURNED	12-Aug-2020

GENERAL COMMENTS	I like to convey my thanks to the Authors for their efforts. In the ABSTRACT section: 'HAND displayed good accuracy for detecting definite RHD only and modest accuracy for detecting any RHD but demonstrated poor accuracy for the detection of borderline RHD alone.' has been stated. But in the RESULTS SECTION, parameters relating to accuracy for 'any RHD' has not been mentioned. If this is mentioned, the ABSTRACT could be more independent.
---

REVIEWER	Bruno Ramos Nascimento Hospital das Clínicas da Universidade Federal de Minas Gerais, Brazil
REVIEW RETURNED	07-Aug-2020

GENERAL COMMENTS	Thank you for addressing some points highlighted in my review. I have no additional concerns.
---

VERSION 2 – AUTHOR RESPONSE

Thank you for your decision and final comments.

In response to reviewer 2 we would like to acknowledge that we have included parameters relating to the accuracy of HAND for 'any RHD' detection in the second sentence of the RESULTS section, as follows; "For any RHD detection, the pooled results from six studies were; sensitivity: 81.6% (95% CI: 76.5%-86.6%) and specificity: 89.8% (84.5%-95.0%)". For this reason we have made no further changes to the manuscript.